# Genomic Analysis of Glycosyltransferases Responsible for Galactose-α-1,3-Galactose Epitopes in *Streptococcus pneumoniae*: Implications for Broadly Protective Vaccination Strategy

**DOI:** 10.3390/vaccines13111148

**Published:** 2025-11-10

**Authors:** Xinjia Mai, Nian Wang, Chenxi Zhu, Yue Ma, Zhongrui Ma, Lan Yin, Dapeng Zhou

**Affiliations:** 1Key Laboratory of Pathogen and Host-Interactions, Department of Immunology and Pathogen Biology, Ministry of Education, Tongji University School of Medicine, 500 Zhennan Road, Shanghai 200331, China; 2211439@tongji.edu.cn (X.M.); 2332219@tongji.edu.cn (N.W.); zhuchenxi@tongji.edu.cn (C.Z.); mayue_2022@tongji.edu.cn (Y.M.); yingmsn@163.com (L.Y.); 2School of Clinical and Basic Medical Sciences, Shandong First Medical University & Shandong Academy of Medical Sciences, Jinan 250117, China; mazhongrui@sdfmu.edu.cn

**Keywords:** α1,3 galactose, *Streptococcus pneumoniae*, 23-valent *Streptococcus pneumoniae* polysaccharide vaccine, anti-Gal antibodies

## Abstract

Background: The origin of natural anti-galactose-α-1,3-galactose (anti-Gal) antibodies in humans is only partially understood. The gut microbiome has been proposed as an important source of galactose-α-1,3-galactose (αGal) epitopes that drive the maturation of anti-Gal–reactive B cells. Certain bacteria expressing αGal epitopes, notably *Escherichia coli* O86:B7, have been shown to elicit anti-Gal antibody responses in α1,3-galactosyltransferase knockout (α3GalT1 KO) mice. In this study, we investigated the interaction between currently widely used bacteria polysaccharide vaccine, the 23-valent pneumococcal polysaccharide vaccine (PPV23), which contains capsular polysaccharides (CPS) from multiple *Streptococcus pneumoniae* serotypes, and host anti-Gal antibodies. Methods: We conducted a genomic analysis to identify α1,3-galactosyltransferase (α3GalT1) genes in *S. pneumoniae* strains. Binding of PPV23 to anti-Gal monoclonal antibodies was evaluated by ELISA, and αGal epitope content in PPV23 was estimated using a four-parameter logistic (4PL) model fitted to the ELISA calibration data. To assess in vivo immunogenicity, we immunized α3GalT1 KO mice with PPV23 and measured serum anti-Gal IgG and IgM titers before and after vaccination. Results: Genomic analysis revealed the presence of α3GalT1 genes in *S. pneumoniae* strains. PPV23 showed specific binding to anti-Gal monoclonal antibodies as detected by ELISA. Quantitative modeling indicated that αGal epitopes are present at low abundance within PPV23, consistent with limited expression of αGal among a minority of included serotypes. Immunization of α3GalT1 KO mice with PPV23 induced a significant rise in anti-Gal IgG titers (mean value from 124 to 384), whereas anti-Gal IgM titers remained relatively unchanged (mean value at the range of 6500–7500). High baseline anti-Gal IgM levels observed in α3GalT1 KO mice are consistent with age-dependent induction by the gut microbiota. Conclusions: These results provide genetic and immunological evidence that αGal epitopes derived from *S. pneumoniae* are present in PPV23 and can engage pre-existing anti-Gal antibodies. Our findings underscore a complex interplay among bacterial glycosyltransferase genes, vaccine polysaccharide composition, and host anti-Gal antibody repertoires, which may influence vaccine immunogenicity. Consideration of host natural antibody profiles may therefore be important for interpreting responses to carbohydrate-based vaccines and for guiding vaccine design.

## 1. Introduction

*Streptococcus pneumoniae* is a Gram-positive extracellular opportunistic pathogen that colonizes the mucosal surfaces of the human upper respiratory tract [1]. This bacterium serves as a major causative agent of a wide spectrum of diseases, ranging from localized infections to severe, life-threatening conditions such as pneumonia, bacteremia, and meningitis [2,3]. Notably, children, the elderly, and immunocompromised individuals exhibit heightened susceptibility to pneumococcal infection [2].

*S*. *pneumoniae* demonstrates remarkable genetic and phenotypic diversity, largely attributable to structural variations in the capsular polysaccharides (CPS). These immunodominant surface structures function both as critical virulence determinants and as the basis for serological classification, with over 100 distinct serotypes currently identified [1,4]. Among these, 23 serotypes account for 80–90% of invasive pneumococcal infections worldwide [5,6].

Since their introduction in the 1970s, pneumococcal polysaccharide vaccines (PPVs) have expanded from formulations covering 6 serotypes to the current 23-valent vaccine (PPV23), which remains the most widely used pneumococcal vaccine globally. Despite its broad coverage and licensure in most countries, PPV23 demonstrates limited effectiveness in two key vulnerable populations: the elderly, due to age-related immunosenescence, and children under 2 years, whose immature immune systems cannot mount adequate responses to polysaccharide antigens [7].

PPV23 contains capsular polysaccharide antigens of serotypes 1, 2, 3, 4, 5, 6B, 7F, 8, 9N, 9V, 10A, 11A, 12F, 14, 15B, 17F, 18C, 19A, 19F, 20, 22F, 23F, and 33F. These serotypes cover approximately 63% to 79% of isolates associated with invasive pneumococcal disease (IPD) in China [8]. A recent systematic review and meta-analysis reported that the predominant serotypes of *S. pneumoniae* in China between 2019 and 2023 were 19F, 6B, 19A and 23F—all of which are included in the protective spectrum of PPV23 [9]. Pneumococcal capsular polysaccharides are thymus-independent antigens, which elicit antibody responses primarily against linear epitopes formed by their repetitive units. In the absence of T-cell help, the response is dominated by short-lived IgM and IgG2 antibodies and fails to establish long-term immune memory [10]. These characteristics contribute to the poor immunogenicity of PPV23 in infants [11]. Although PPV23 was widely recommended for the elderly population, studies have reported non-significant protection against IPD in healthy older adults (65% incidence reduction, OR 0.35; 95% CI 0.08–1.49), with minimal efficacy observed among high-risk elderly individuals [12].

Human endogenous anti-carbohydrate antibodies play essential roles in host defense by targeting microbial glycans—including bacterial, fungal, and other microbial carbohydrates—thereby preventing systemic infections and maintaining microbiome homeostasis [13,14].

Anti-Gal antibodies are natural antibodies present in unusually high amounts in human sera, with distinct specificity for αGal epitopes. Anti-Gal antibodies have been shown to interact with various strains of *Escherichia coli*, *Klebsiella*, and *Salmonella*, including *E. coli* O86, *K. pneumoniae* 18033, and *S. minnesota* [15,16]. However, it remains unclear whether all these bacteria express genes responsible for synthesizing the αGal epitopes.

In healthy individuals, anti-Gal antibody levels vary across populations, with lower concentrations typically observed in infants and the elderly. These antibodies play a crucial role in host defense by binding diverse pathogens [17], activating the classical complement pathway, and promoting phagocytosis [18,19]. Enhancing anti-Gal antibody levels, therefore, represents a potential therapeutic strategy.

The anti-Gal antibodies also exhibit polyreactivity, binding to multiple enteric bacterial strains, including those lacking α1,3-galactosyltransferase (α3GalT1)-encoding genes [20,21]. Polyreactive antibodies are defined by their biologically relevant affinities for at least two distinct epitopes [22,23]. Such polyreactivity may be important for first-line defense against invading pathogens. Conversely, studies in α3GalT1 KO mice have shown that gut microbiota can induce a broad spectrum of natural polyreactive anti-glycan antibodies, including low titers of anti-Gal antibodies [24].

In 2021, Jens Magnus Bernth Jensen et al. [19] demonstrated that anti-Gal IgG antibodies isolated and purified from human serum exhibited specific binding to 48 serotypes of *S*. *pneumoniae* by flow cytometry. Notably, these reactive serotypes (9V, 19A, 6B, 7F, 10A, 12F, and 33F) include major immunogenic components of PPV23. These findings suggest that PPV23, which contains capsular polysaccharide antigens from 23 pneumococcal strains, may induce an anti-Gal antibody-mediated immune response in humans, including complement activation. Importantly, the magnitude of this vaccine-induced response appears to depend on endogenous anti-Gal antibody titers [25].

In this study, we conducted genomic analysis of the 23 *S. pneumoniae* serotypes included in PPV23 to identify potential α3GalT1 genes. We further evaluated the binding of PPV23 to several anti-Gal monoclonal antibodies and assessed the immunogenicity of PPV23 in α3GalT1 KO mice. By comparing anti-Gal antibody levels before and after immunization, our work aims to clarify whether PPV23 can serve as a novel inducer of anti-Gal antibodies. This study provides new insights into the role of pneumococcal vaccines in modulating natural anti-carbohydrate immunity and highlights a previously unrecognized immunological function of PPV23.

## 2. Materials and Methods

### 2.1. Mice

All mouse experiments were performed in strict accordance with the “Guiding Principles in the Use and Care of Animals” published by the National Institutes of Health (NIH Publication No. 85-23, Revised 1996) and were approved by the Institutional Animal Care and Use Committee of Tongji University. The α3GalT1 KO C57BL/6 mice were purchased from the Saiye Biotech Limited Company (Guangzhou, China) and maintained under specific pathogen-free (SPF) conditions. Mice were bred in our SPF facility, and 12- to 14-week-old animals of both sexes were used for experiments. Animals were randomly allocated to treatment and control groups to minimize bias. Throughout the study, all animals were handled and sampled under consistent experimental conditions, and no data points or animals were excluded from the analysis. No adverse events were observed during the course of the study. No specific humane endpoints were established for this study, as the procedures were not expected to induce significant pain or distress beyond momentary discomfort.

### 2.2. Immunization

Mice (*n* = 10) were immunized intraperitoneally with 5 μg of 23-valent pneumococcal polysaccharide vaccine (Pneumovax 23; Merck & Co., Whitehouse Station, NJ, USA) combined with 50 μg of PIKA adjuvant (Yisheng Biopharma Ltd., Beijing, China). Control mice received equal saline supplemented with 50 μg of PIKA adjuvant. Immunizations were performed on days 0, 7, and 14. Serum samples were collected before the first immunization and on day 21, and stored at −20 °C until analysis.

### 2.3. α3GalT1 Gene Sequences and tBLASTN-Based Homology Screening

We used published mammalian α3GalT1 genes to search for gene analogs, including mouse α3GalT1 [26] (Gene ID: 14594), *WCIN* [27] (GeneBank: AB795223.1), *WbnI* [28] (GenBank: CAK7542307.1), *GlyE* [29] (GenBank: AP026930.1).

The capsular polysaccharide (cps) biosynthetic gene sequences for the 90 known serotypes of *S. pneumoniae* (accession numbers CR931632–CR931722) were retrieved from the GenBank database (http://www.ncbi.nlm.nih.gov/Genbank; accessed on 21 March 2025) [30]. The protein sequences of the three genes of interest were used as queries in a tBLASTN (BLAST+ 2.15.0) search against the downloaded cps gene dataset. Hits were considered significant based on the following thresholds: identity ≥ 20%, e-value ≤ 10, and query coverage ≥ 5%.

### 2.4. Monoclonal Antibodies

The protein sequences of Gal-13 mAb [31] (VH: AAG02035.1; VL: AAG02029.1), 15.101 mAb [32] (VH: AAK69385.1 and VL sequence from reference [33]), and M86 mAb [34] (VH:2260471759; VL:2260471758) were retrieved from the NCBI. Gal-13 mAb and 15.101 mAb were expressed as mouse IgG1 using a vector provided by Sino Biological Ltd. (Beijing, China), while M86 was expressed as mouse IgM.

### 2.5. Detection of αGal Antigen of PPV23 by ELISA

PPV23 or BSA-αGal (Dextra Laboratories, Reading, UK) was diluted to 1 μg/mL in 0.01% NaN3/PBS, and 100 μL/well was coated (in duplicate) on 96-well plates (Corning, Inc., Corning, NY, USA) and incubated at 37 °C for 5 h. After washing with 1% Brij-35 (Sigma, St. Louis, MO, USA)/TBS, blocking was performed by using 1% BSA/PBS for 1 h. After blocking, M86, Gal-13 and 15.101 antibodies were added in 2- or 5-fold serial dilution in 1% BSA/PBS.

After incubation for 2 h at 37 °C, goat-anti-mouse-IgM-HRP or goat-anti-mouse-IgG-HRP (1:5000; SouthernBiotech, Birmingham, AL, USA) was added at room temperature for 1 h. Following the addition of 100 μL per well of TMB substrate (Biolegend, San Diego, CA, USA), the reaction was stopped with 100 μL per well of 4% sulfuric acid. Plates were read at 450 nm using a Spark microplate reader (Tecan, Männedorf, Switzerland).

### 2.6. Competitive ELISA with Galα1-3Gal Disaccharide

Solutions and conditions were the same as for ELISA, as described above. Plates were coated with 100 µL/well of 1 µg/mL of PPV23 or BSA-αGal and blocked as described above.

Galα1-3Gal disaccharide (Dextra Laboratories, Reading, UK) was 2-fold serially diluted from 0.5 mg/mL (1.46 mM), and mixed with 1 μg/mL (1.12 nM) of M86 antibody in a 1:1 volume ratio. After incubation at 37 °C for 2 h, the mixture was added to an antigen-coated plate and incubated for 2 h at 37 °C. Then, the secondary antibody (goat-anti-mouse-IgM-HRP, SouthernBiotech, Birmingham, AL, USA) was added (1:5000) at room temperature for 1 h. The HRP activity was detected as above.

### 2.7. Quantification of the αGal Epitope of PPV23 Using BSA-αGal as Standard

BSA-αGal was serially diluted two-fold from 0.05 μg/mL in triplicate to generate the standard curve. PPV23 was coated at concentrations of 5 μg/mL, 10 μg/mL, and 20 μg/mL. The plates were then incubated at 37 °C for 5 h. After washing with 1% Brij-35 in TBS, blocking was performed using 1% BSA in PBS for 1 h. Next, 0.1 μg/mL of the M86 antibody (diluted in 1% BSA/PBS) was added and incubated at 37 °C for 2 h. A secondary antibody (goat anti-mouse IgM-HRP, SouthernBiotech, Birmingham, AL, USA) was applied at a 1:5000 dilution and incubated at room temperature for 1 h. Finally, the HRP was detected as described above.

### 2.8. Detection of Anti-Gal and Anti-PPS Antibodies in Mouse Serum

Anti-Gal and anti-PPS antibodies were measured by coating 96-well plates (Corning, Inc., Corning, NY, USA) with either 100 µL of PPV23 or BSA-αGal (1 µg/mL) in PBS at 37 °C for 5 h. All plates were washed with 1% Brij-35 (Sigma, St. Louis, MO, USA)/TBS, and blocked with 1% BSA in PBS for 2 h at 37 °C. Sera samples were serially diluted two- or five-fold in PBS, and incubated overnight at 4 °C, followed by the addition of goat anti-mouse-IgM-HRP or goat anti-mouse-IgG-HRP (1:5000; SouthernBiotech, Birmingham, AL, USA) for 1 h of incubation at room temperature. The HRP was detected as above.

### 2.9. Statistical Analysis

Analyses were conducted using GraphPad Prism 9.0. The statistical differences in sera reactivity against αGal and PPS were evaluated using the Wilcoxon signed-rank test in the GraphPad Prism 9 program. Continuous variables are presented as the mean ± SEM.

## 3. Result

### 3.1. S. pneumoniae Expresses Similar Sequences to α3GalT1

We used published mammalian and bacterial α1,3 glycosyltransferase genes to search for analog genes. The *WCIN* gene of *S. pneumoniae* has been previously verified to synthesize Galα(1-3)Glcα-PP-lipid [27]. GlyE protein has been reported to possess a typical glycosyltransferase family 8 (GT8) domain and plays a crucial role in the polymorphic O-type glycosylation of PsrP [29] in *S. pneumoniae*. This enzyme catalyzes not only the third step of PsrP glycosylation: transferring galactose from UDP-galactose to the terminal glucose residue of an already glycosylated PsrP (utilizing the short substrate PsrP-GlcNAc-Glc), but also facilitates the fourth step in this process. The sugar moiety from UDP-galactose can be transferred to the terminal sugar unit of PsrP-GlcNAc-Glc-Gal, thereby resulting in the formation of a Galα1-3Glc structure [35]. We also searched for the *WbnI* gene in *E. coli* O86, a strain widely recognized as an αGal-positive bacterium. The *WbnI* gene encodes a galactosyltransferase responsible for introducing α-(1→3)-Galp residues as side chains of the polysaccharide [36].

We performed BLAST analysis (v. 2.15.0) of these three gene sequences across 93 serotypes of *S. pneumoniae* [37] (Table 1). The results revealed greater than 90% sequence similarity between the *WCIN* genes of serotypes 6A and 33D and the known *WCIN* sequences. It is worth noting that the capsular polysaccharide of serotype 6A is one of the components of PPV23. In contrast, the *WbnI* and *GlyE* genes showed relatively low conservation across different serotypes, suggesting potential phylogenetic divergence among strains. These findings provide indirect evidence supporting the polyreactivity of anti-Gal antibodies.

In summary, we have identified *WCIN* as the key gene encoding α3GalT1 in *S. pneumoniae* through BLAST analysis. However, this gene is not expressed in all serotypes. Notably, serotype 6A, which exhibits high *WCIN* expression, is also a component of PPV23. This finding warrants further investigation.

### 3.2. Binding of PPV23 to Anti-Gal Antibodies and Inhibition of Binding by Galα1-3Gal Disaccharide

To examine whether PPV23 capsular polysaccharides contain Galα1-3Gal-like disaccharide motif, we performed an ELISA to detect interactions between PPV23 and monoclonal anti-Gal antibodies. We employed BSA-αGal as a positive control for three anti-Gal monoclonal antibodies: M86 mAb, 15.101 mAb, and Gal-13 mAb [31,32,33,34].

The ELISA results (Figure 1A) revealed differential binding affinities of PPV23 toward the three antibodies, with the strongest interaction observed for M86 mAb and the weakest for Gal-13 mAb. A similar pattern was observed for BSA-αGal binding, where M86 exhibited the highest affinity and Gal-13 the lowest (Figure 1B). Competitive binding assays using the Galα1-3Gal disaccharide demonstrated dose-dependent inhibition, reaching maximal inhibitory effect (approximately 47%) at 0.73 mM for the M86 mAb-PPV23 interaction (Figure 1C) [38]. The dose-dependent inhibition by free Galα1-3Gal disaccharide further supports the specificity of this interaction. These results provide direct evidence that the Galα1-3Gal epitope represents the predominant binding motif for anti-Gal antibodies on the pneumococcal polysaccharide. The differential binding profiles among monoclonal antibodies underscore the functional diversity of the natural anti-Gal response.

### 3.3. Quantification of αGal Epitopes in PPV23

To examine the quantity of Galα1-3Gal-like disaccharide, quantification of αGal epitopes in PPV23 vaccine was performed using ELISA with M86 mAb [39]. A standard curve was generated using serial dilutions of BSA-αGal (ranging from 4.88 × 10^−5^ to 0.05 μg/mL) (Figure 2). The binding data were analyzed using a four-parameter logistic (4PL) regression model to generate the standard curve. The model equation was defined as: Y = Bottom + (Top − Bottom)/(1 + 10^((LogEC_50_ − X) × HillSlope)), where Y represents the observed OD value, and X is the logarithm of the concentration (Figure 2A). ELISA revealed that the OD value of 5 μg/mL PPV23 was too low to determine the corresponding BSA-αGal concentration. In contrast, the OD value of 10 μg/mL PPV23 was equivalent to that of BSA-αGal at 4.39 × 10^−4^ μg/mL, while the OD value of 20 μg/mL PPV23 was equivalent to BSA-αGal at 5.46 × 10^−4^ μg/mL (Figure 2B,C). This non-linear relationship between PPV23 concentration and αGal equivalency suggests potential epitope masking or steric hindrance at higher polysaccharide densities.

Due to the non-linear relationship between PPV23 concentration and αGal equivalency, the 10 μg/mL data point was used as the reference for quantitative estimation. Based on this calculation, 1 μg/mL of PPV23 contains αGal epitopes equivalent to 4.39 × 10^−5^ μg/mL of the BSA-αGal standard. Given the molecular characteristics of BSA-αGal (MW: 86.14 kDa; 33 αGal residues per molecule), this concentration corresponds to 4.65 × 10^−4^ pmol/mL, representing 2.79 × 10^8^ BSA-αGal molecules and 9.21 × 10^9^ total αGal epitopes per milliliter.

For PPV23 capsular polysaccharides, a representative molecular weight of 507.4 kDa was used for calculation, which corresponds to the arithmetic mean of the reported minimum molecular weight specifications of its 23 capsular polysaccharides [40]. Since PPV23 contains 23 distinct polysaccharides with heterogeneous molecular sizes, this value should be considered an approximation. Based on this estimate, 1 μg/mL of PPV23 equates to 1.97 × 10^−12^ mol (1.19 × 10^12^ molecules) of polysaccharide. These calculations indicate that PPV23 polysaccharides display approximately 9.21 × 10^9^ αGal epitopes in total, corresponding to an average density of 0.008 αGal motifs per polysaccharide molecule. This low epitope density likely results from the limited expression of αGal among a minority of serotypes in PPV23, such as serotype 6A with the *WCIN* gene. Since the calculation treated PPV23 as a composite of all 23 capsular polysaccharides, the average αGal content was substantially diluted by non- or low-expressing serotypes. Thus, not all serotypes contribute equally to anti-Gal binding, underscoring serotype-specific variation in αGal epitope expression.

### 3.4. The PPV23 Immunization Induces Anti-Gal IgG Antibody Production in α3GalT1 Knockout Mice

To examine the immunogenicity of Galα1-3Gal-like structure, we immunized α3GalT1 KO mice with PPV23. The study employed three experimental groups: α3GalT1 KO mice intraperitoneally immunized with PPV23 (experimental group); α3GalT1 KO mice injected with saline (negative control); and wild-type (WT) mice immunized with PPV23 intraperitoneally (WT mice control) (Figure 3).

Serum antibody levels quantified by ELISA revealed successful induction of anti-pneumococcal polysaccharide (PPS)-specific antibodies in both PPV23-immunized α3GalT1 KO and WT mice, with IgM (mean titer = 42,500 in both groups) and IgG (mean titer = 1414 in α3GalT1 KO; 904 in WT) responses significantly elevated compared to saline-treated α3GalT1 KO controls. Notably, no significant differences (*p* > 0.05) were observed in anti-PPS IgM or IgG titers between immunized α3GalT1 KO and WT groups (Figure 3A,B), indicating that α3GalT1 gene knockout did not substantially alter humoral immune responses to PPS antigens in PPV23.

In PPV23-immunized α3GalT1 KO mice, serum anti-Gal IgG levels exhibited a significant increase relative to both pre-immunization baselines and saline-treated controls (*p* < 0.05). In contrast, anti-Gal IgM titers showed no statistically significant changes post-immunization (Figure 3D). These results demonstrate that PPV23 immunization effectively elicits anti-Gal-specific IgG antibody responses in α3GalT1 KO mice. While anti-PPS IgG titers tended to be higher in KO mice compared with WT controls (mean titer = 1414 in α3GalT1 KO vs. 904 in WT), this difference did not reach statistical significance. This outcome may reflect the substantially lower concentration of anti-Gal antibodies in mouse serum compared with humans [24,41].

## 4. Discussion

We conducted a comprehensive analysis of the genes responsible for the αGal epitope biosynthesis in *S. pneumoniae* and validated the presence of these epitopes in PPV23 using ELISA. Furthermore, immunization of α3GalT1 knockout mice with PPV23 induced robust anti-Gal IgG antibody production. Thus, we provide genetic and immunological evidence that PPV23 is immunogenic in an animal host deficient in the αGal epitope.

Our study reveals a fundamentally new mechanism. For the first time, we demonstrate that immunization with purified bacterial capsular polysaccharides from the clinically relevant PPV23 is sufficient to induce a significant anti-Gal antibody response. This finding is highly significant as it shifts the paradigm from live bacteria to a defined vaccine preparation, highlighting the critical role of anti-Gal antibodies in the immunogenicity of carbohydrate-based vaccines. Our work thus provides a direct immunological link between a widely used human vaccine and the natural anti-Gal antibody system, opening new avenues for rational vaccine design.

Indeed, the α3GalT1 KO mouse, initially developed for xenotransplantation rejection research [42,43], has subsequently been utilized—due to the unique immunological characteristics of the αGal epitope and anti-Gal antibodies—in studies spanning anti-tumor immunity [44] and experimental Chagas disease models [45]. These collective studies have demonstrated that, aside from the absence of αGal epitopes, these mice do not exhibit major alterations, in general, in immune function or development. Therefore, the observed increase in anti-Gal IgG following PPV23 immunization is most directly attributable to the absence of endogenous αGal and the subsequent immune response to bacterial epitopes, rather than to any generalized immune dysfunction. Further investigation is required to elucidate the specific mechanisms underlying this immune response and its potential beneficial effects on the host.

The PPV23 preparation exhibits a total polysaccharide concentration of 4.28 nM per vial. In healthy adults, serum anti-Gal IgG concentrations average approximately 133 μM (MW = 150 kDa), with interindividual variation of up to 400-fold. Given this substantial antibody excess, which dominates the binding equilibrium, combined with the established affinity of anti-Gal for αGal epitopes, our analysis predicts that a significant proportion of polysaccharide molecules will form immune complexes with anti-Gal antibodies following vaccine administration in humans. This molecular interaction provides a mechanistic basis for vaccine-induced complement activation and offers a plausible explanation for the observed heterogeneity in immune responses among vaccine recipients.

Our study elucidates the interaction between anti-Gal antibodies and the PPV23, thereby providing both genetic and immunological evidence for its immunogenicity in α3GalT1-deficient hosts. This finding expands the understanding of how pre-existing natural antibodies shape vaccine responses. By revealing the dual role of vaccine polysaccharides as both antigens and ligands for abundant natural antibodies, our work underscores the importance of considering host antibody repertoires in vaccine evaluation and design, and it opens new avenues for optimizing carbohydrate-based vaccines to achieve more consistent and effective protection.

However, several limitations should be acknowledged. In our study, we could not observe significant changes in anti-Gal IgM titers in α3GalT1 KO mice before and after PPV23 immunization. This might be because the fact the cross-reactivity of natural anti-glycan antibodies is very broad, and does not distinguish the low abundance of αGal epitopes in PPV from other highly abundant glycan structures. In the IgG response, high-affinity, antigen-specific anti-Gal B cells were selected, whereas the IgM response arose without selection of high-affinity B cells.

The other limitation of our study is that the titer of natural anti-Gal antibodies in the α3GalT1 KO model is much lower than in humans. It remains to be studied how to further boost such antibodies naturally. Notwithstanding this constraint, our findings remain mechanistically informative. The significant anti-Gal IgG response induced by PPV23 in this model, despite the suboptimal baseline, demonstrates that the vaccine polysaccharides can effectively engage the available anti-Gal repertoire to initiate a specific immune response. However, the induced titer is considerably lower than basal levels in humans [41], underscoring that PPV23 alone is likely insufficient to recapitulate human antibody levels, where additional factors like chronic dietary exposure to αGal epitopes may be critical. Therefore, directly investigating whether PPV23 can modulate pre-existing anti-Gal antibodies in vaccinated adults represents a critical next step, and our work provides a solid foundation for such clinical studies.

Our data suggest that bacterial capsular polysaccharides expressing αGal alone are not sufficient to induce high titers of anti-Gal IgM and IgG antibodies. The elevated anti-Gal antibody levels in humans may instead be driven by additional stimuli, such as dietary intake of Galα1-3Gal disaccharide-containing foods or colonization by αGal-positive bacteria such as *E. coli* O86:B7. For example, Yimaz et al. [46] first colonized α3GalT1 KO mice with *E. coli* O86:B7 to induce a high titer of anti-Gal antibodies, and further demonstrated the potent protective efficacy in malaria infection by induced antibodies. It also remains to be studied whether PPV23-induced anti-Gal antibodies are produced by a different population of B cells than anti-Gal antibodies induced by *E. coli* O86:B7 or Galα1-3Gal disaccharide-containing foods.

## 5. Conclusions

The purpose of this study was to investigate the expression of *α3GalT1* in *S*. *pneumoniae* through BLAST analysis and to characterize the interaction between PPV23 and monoclonal anti-Gal antibodies. We also established a quantitative assay for the detection of αGal epitopes in PPV23 and demonstrated that PPV23 immunization induced anti-Gal antibody production in α3GalT1 KO mice. Collectively, these findings suggest a potential synergistic interaction between naturally occurring anti-Gal antibodies and vaccine-induced anti-pneumococcal polysaccharide antibodies, thereby providing new insights into the mechanisms of PPV23-induced immunity in humans.

## Figures and Tables

**Figure 1 vaccines-13-01148-f001:**
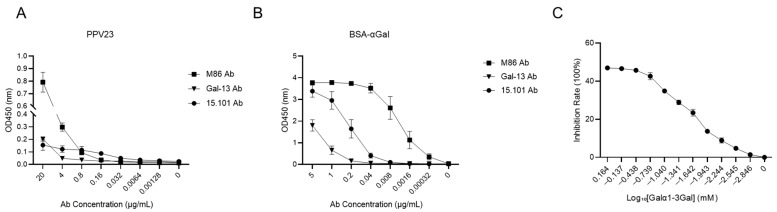
PPV23 can bind to anti-Gal antibodies and be competitively inhibited by Galα1-3Gal. (**A**) ELISA showing binding of PPV23 to the anti-Gal monoclonal antibodies M86, 15.101, and Gal-13 mAb. (**B**) ELISA analysis of BSA-αGal binding to the anti-Gal monoclonal antibodies M86, 15.101, and Gal-13 mAb. (**C**) Competitive ELISA demonstrating dose-dependent inhibition of M86 mAb binding to PPV23 by Galα1-3Gal disaccharide. Data are representative of three independent experiments, and values are shown as mean ± SEM.

**Figure 2 vaccines-13-01148-f002:**
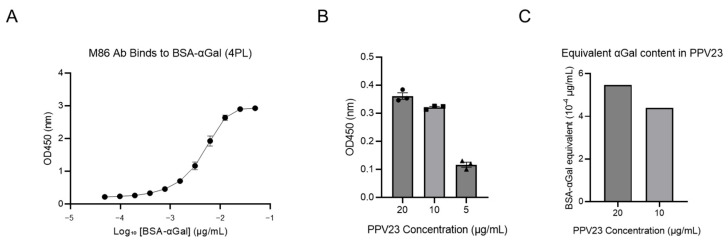
Quantification of αGal epitopes in PPV23. (**A**) Four-parameter logistic (4PL) non-linear regression of the BSA-αGal standard curve when binding to M86 mAb. The solid line indicates the best-fit model defined as Y = Bottom + (Top − Bottom)/(1 + 10^((LogEC_50_ − X) × HillSlope)). Best-fit parameters: EC_50_ = 4.71 nM (95% CI: 4.14–5.40 nM), HillSlope = 1.61, Top = 3.04, Bottom = 0.25, R^2^ = 0.99. Data points represent mean ± SEM. (**B**) The corresponding OD values when different doses of PPV23 bind to M86 mAb. (**C**) The BSA-αGal concentrations corresponding to different doses of PPV23, calculated based on the equation derived from panel A. The OD_450_ value for 5 μg/mL PPV23 fell below the lower limit of the equation and is therefore not shown.

**Figure 3 vaccines-13-01148-f003:**
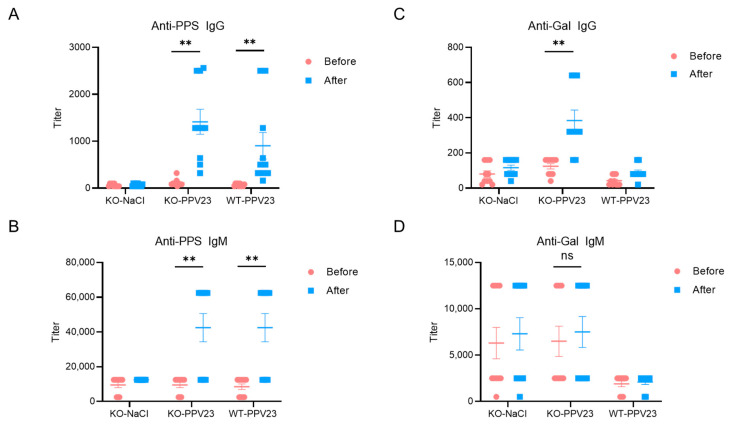
Anti-PPS and anti-Gal antibody responses in WT and α3GalT1 KO mice following PPV23 immunization. Serum levels of (**A**) PPS-specific IgG, (**B**) PPS-specific IgM, (**C**) αGal-specific IgG, and (**D**) αGal-specific IgM were measured by ELISA. Data were pooled from two independent experiments (*n* = 10 mice per group). Values are presented as mean ± SEM. Each symbol represents an individual mouse. Statistical significance was determined using the Wilcoxon signed-rank test; ** *p* < 0.01, ns (not significant).

**Table 1 vaccines-13-01148-t001:** Representative BLAST results of *WCIN*, *WbnI*, and *GlyE* queries against the *S. pneumoniae* pan-genome.

Query	Subject	Identity	Match_length
*WCIN*	**SPC06A_0009|wciN**	99.682	314
*WCIN*	SPC33D_0009|wciN	92.357	314
*WCIN*	**SPC06A_00006|wciN**	91.72	314
*WCIN*	SPC33C_0009|wciN	64.331	314
*WCIN*	SPC33B_0009|wciN	61.146	314
*WCIN*	**SPC01_0014|gla**	29.412	51
*WCIN*	**SPC06B_00003|wzd**	40	30
*WCIN*	SPC23B_0006|wzd	40	30
*WCIN*	**SPC11A_0005|wzd**	40	30
*WbnI*	SPC10C_0013|wcrD	29.825	57
*WbnI*	SPC10F_0013|wcrD	29.825	57
*WbnI*	SPC06C_11|rmlA	45.833	24
*WbnI*	SPC34_0009|wciB	30.357	56
*GlyE*	SPC08_0017|HG266	66.667	12

Homologous sequences matching the *WCIN*, *WbnI*, and *GlyE* query sequences are listed, with percent identity and alignment length indicated. Subject sequences derived from serotypes included in PPV23 are highlighted in bold.

## Data Availability

All data generated or analyzed during this study are included in this published article (figures and tables).

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
