# Peer review of "Genomic Analysis of Glycosyltransferases Responsible for Galactose-α-1,3-Galactose Epitopes in Streptococcus pneumoniae: Implications for Broadly Protective Vaccination Strategy"

_vaccines, 2025, doi:10.3390/vaccines13111148_

Round 1
Reviewer 1 Report
Comments and Suggestions for Authors
Attached.

Author Response
- Are any biological functions of the mice affected by the gene knockout? Could any of these functions modify the immune response?
We thank the reviewer for raising this important point regarding the animal model. Indeed, the C57BL/6 α-1,3-galactosyltransferase knockout (α3GalT1 KO) mouse, initially developed for xenotransplantation rejection research1,2, has subsequently been utilized—due to the unique immunological characteristics of the αGal epitope and anti-Gal antibodies—in studies spanning antitumor immunity3 and experimental Chagas disease models4. These collective studies have demonstrated that, aside from the absence of αGal epitopes, these mice do not exhibit major alterations in general immune function or development. Therefore, the observed increase in anti-Gal IgG following PPV23 immunization is most directly attributable to the absence of endogenous αGal and the subsequent immune response to bacterial epitopes, rather than to any generalized immune dysfunction. We have added this paragraph in discussion (Page 11 Line 349 to 357)
- Anti-Gal antibody titers are lower in the young and elderly in humans (Intro), but significant in adults. Assuming this translates well in mice (which is uncertain), this would imply that wild-type adult mice (14-weeks old) should react to PPV23. Could be interesting to study the effect of age (with mice or, if infeasible due to species limitations, human serum) on anti-Gal antibodies after PPV23. Naturally, going clinical would be far beyond the scope of this work. A rationale for using adult mice would however be welcome.
We thank the reviewer for this insightful comment regarding the potential influence of age. Our decision to use 14-week-old adult mice was based on two key considerations. First, it is well-established in mice that anti-Gal antibody titers are shaped by the gut microbiota, which undergoes dynamic colonization until approximately 3 months of age, after which both the microbiota and the anti-Gal titers stabilize5. Using adult mice thus ensures a stable baseline for the anti-Gal antibody pool, minimizing variability from ongoing microbial colonization. Second, since our primary goal was to investigate the specific impact of anti-Gal antibodies on the anti-polysaccharide (anti-PPS) response, we needed to control for the known confounding effect of age on the response to PPV23 itself. Using immunologically mature adult mice, which mount a robust and consistent anti-PPS response analogous to healthy human adults, allowed us to isolate the variable of interest more clearly.
- The study shows that anti-Gal IgG cannot be induced in healthy adult mice. On top of this, the epitope has a fairly low density (0.08/polysaccharide). Is there evidence that PPV23 creates antiGal antibodies in adults?
We thank the reviewer for this insightful comment. We fully acknowledge the limitations highlighted by the reviewer, namely the lower endogenous anti-Gal titers and less potent complement activity in mice compared to humans. Precisely because of these known species differences, we believe the significant anti-Gal IgG response observed even in our mouse model is mechanistically informative. It suggests that the vaccine polysaccharides can effectively engage the available anti-Gal antibodies.
On the other hand, the anti-Gal IgG titer induced by PPV23 in α3GalT1 KO mice is much lower than human adults6. We can not exclude the possibility that alpha1,3 galactose of other origin (such as food origin) might play significant roles in inducing anti-Gal IgG antibodies in human.
We agree that directly investigating whether PPV23 can modulate or boost anti-Gal antibodies in vaccinated adults is a critical and logical next step. Our current findings provide a solid foundation and a clear rationale for such future clinical studies, which are now a primary objective of our ongoing research program. We have added this paragraph in discussion (Page 12 Line 386 to 395)
4.The abstract describes background and methods, but does not explain the goal of studying anti-Gal antibodies in PPV23.
We thank the reviewer for this suggestion. We have incorporated the research objective into the abstract as recommended(Page1, line20).
References:
- Thall, A D et al. “Oocyte Gal alpha 1,3Gal epitopes implicated in sperm adhesion to the zona pellucida glycoprotein ZP3 are not required for fertilization in the mouse.” The Journal of biological chemistry vol. 270,37 (1995): 21437-40. doi:10.1074/jbc.270.37.21437
- Pearse, M J et al. “Anti-xenograft immune responses in alpha 1,3-galactosyltransferase knock-out mice.” Sub-cellular biochemistry vol. 32 (1999): 281-310. doi:10.1007/978-1-4615-4771-6_12
- Posekany, Karla J et al. “Suppression of Lewis lung tumor development in alpha 1,3 galactosyltransferase knock-out mice.” Anticancer research vol. 24,2B (2004): 605-12.
- Ayala, Edward Valencia et al. “C57BL/6 α-1,3-Galactosyltransferase Knockout Mouse as an Animal Model for Experimental Chagas Disease.” ACS infectious diseases vol. 6,7 (2020): 1807-1815. doi:10.1021/acsinfecdis.0c00061
- Bello-Gil, Daniel et al. “The Formation of Glycan-Specific Natural Antibodies Repertoire in GalT-KO Mice Is Determined by Gut Microbiota.” Frontiers in immunology vol. 10 342. 5 Mar. 2019, doi:10.3389/fimmu.2019.00342
- Lee, Eun Jin et al. “Immunoglobulin M and Immunoglobulin G Subclass Distribution of Anti-galactose-Alpha-1,3-Galactose and Anti-N-Glycolylneuraminic Acid Antibodies in Healthy Korean Adults.” Transplantation proceedings vol. 53,5 (2021): 1762-1770. doi:10.1016/j.transproceed.2021.01.01
Reviewer 2 Report
Comments and Suggestions for Authors
The approaches used to identify and characterize the glycosyltransferases are generally sound and could be significant for development of broad-spectrum vaccine. Some minor comments are suggested
1-Methods for genomic analysis need to be added for alpha-1,3-galactosyltransferase genes in Streptococcus pneumoniae strains.
2-Conc of stopped solution—sulfuric acid needs to be added.
3-How the antibody titer of this vaccine is significant as compared to previously known vaccines
4-The bacteria name should be italic, and it needs to be checked in the whole manuscript.
Author Response
1-Methods for genomic analysis need to be added for alpha-1,3-galactosyltransferase genes in Streptococcus pneumoniae strains.
A:We thank you for this suggestion. The relevant information has been added to the Materials section and can be found on Page 5, Line 170 to 175 of the revised manuscript.
2-Conc of stopped solution—sulfuric acid needs to be added.
A: We agree with the reviewer. The concentration of the stopped solution (sulfuric acid) used was 4%, with a volume of 100 μL per well. This detail has been added to the Methods section on Page 5, Line 190 to 192.
3-How the antibody titer of this vaccine is significant as compared to previously known vaccines
A: We agree with the reviewer that the anti-Gal IgG titer induced by PPV23 is relatively low in a3GalT1 KO mice. Further studies should be performed in human subjects. We also added potential role of other origins of Gala1,3Gal disaccharide in inducing anti-Gal antibodies (Pag 1, Line 402-405).
4-The bacteria name should be italic, and it needs to be checked in the whole manuscript.
A:We thank the reviewer for this reminder. We have now carefully checked the entire manuscript and ensured that all bacterial names are italicized.
Reviewer 3 Report
Comments and Suggestions for Authors
This manuscript addresses the origin of natural anti-galactose-α-1,3-galactose (anti-Gal) antibodies in humans, a question of considerable immunological relevance. The authors propose that bacterial αGal epitopes may contribute to anti-Gal antibody induction. They further demonstrate that the pneumococcal polysaccharide vaccine (PPV23) can enhance anti-Gal IgG levels in α1,3-galactosyltransferase knockout (α3GalT1 KO) mice.
The study is conceptually interesting and presents potentially novel evidence linking a licensed vaccine to anti-Gal induction. However, while the abstract is promising, several aspects require clarification and improvement before the manuscript can be considered for publication.
Major Comments:
The novelty and significance should be clearly distinguished from prior work showing bacterial induction of anti-Gal responses.
Given that many bacterial glycans share structural motifs with αGal, additional controls should be described to ensure antibody specificity (e.g., inhibition assays, glycan arrays). If not feasible, please discuss potential cross-reactivity and specificity.
Ensure consistency in terminology (e.g., αGal vs. α-Gal, α3GalT1 vs. α1,3-galactosyltransferase).
Minor Comments:
After the first mention of Streptococcus pneumoniae, use S. pneumoniae. Always write it in italics (in the title and in the keywords).
Add spacing and typographical consistency for numbers (e.g., “6,500” not “6500”).
Author Response
1.The novelty and significance should be clearly distinguished from prior work showing bacterial induction of anti-Gal responses.
We thank the reviewer for this insightful comment. We have now clarified the novelty and significance of our work in the Discussion section (Page 11, Line 341 to 348).
While prior studies, such as those demonstrating anti-Gal antibody induction in knockout mice via oral administration of live E. coli O86 or the correlation between higher anti-Gal titers in SPF mice and gut microbiota colonization, have established the role of bacteria in stimulating anti-Gal responses,
our study reveals a fundamentally new mechanism. For the first time, we demonstrate that immunization with purified bacterial capsular polysaccharides from the clinically relevant 23-valent pneumococcal vaccine (PPV23) is sufficient to induce a significant anti-Gal antibody response. This finding is highly significant as it shifts the paradigm from live bacteria to a defined vaccine preparation, highlighting the critical role of anti-Gal antibodies in the immunogenicity of carbohydrate-based vaccines. Our work thus provides a direct immunological link between a widely used human vaccine and the natural anti-Gal antibody system, opening new avenues for rational vaccine design.
2.Given that many bacterial glycans share structural motifs with αGal, additional controls should be described to ensure antibody specificity (e.g., inhibition assays, glycan arrays). If not feasible, please discuss potential cross-reactivity and specificity.
We thank reviewer for pointing out the specificity of induced anti-Gal antibodies. Due to limited availability of vast amount of human microbime, we could not perform experiments to examine this. However, Galili et al. and other laboratories have reported that anti-Gal antibodies bind to human flora1,2,3. We have cited these references to highlight the polyreactive nature of anti-Gal antibodies.
3.Ensure consistency in terminology (e.g., αGal vs. α-Gal, α3GalT1 vs. α1,3-galactosyltransferase).
We thank the reviewer for this suggestion. In response, we have standardized the terminology throughout the manuscript, referring to the galactose-α-1,3-galactose as αGal, the anti-galactose-α-1,3-galactose antibody as anti-Gal antibody, and the α1,3-galactosyltransferase as α3GalT1.
Minor Comments:
After the first mention of Streptococcus pneumoniae, use S. pneumoniae. Always write it in italics (in the title and in the keywords).
We thank the reviewer for this comment. The suggested changes have been implemented throughout the manuscript.
Add spacing and typographical consistency for numbers (e.g., “6,500” not “6500”).
We thank the reviewer for this comment. The suggested changes have been implemented throughout the manuscript.
References:
- Galili, U et al. “Interaction between human natural anti-alpha-galactosyl immunoglobulin G and bacteria of the human flora.” Infection and immunity vol. 56,7 (1988): 1730-7. doi:10.1128/iai.56.7.1730-1737.1988
- Bernth Jensen, Jens Magnus et al. “Abundant human anti-Galα3Gal antibodies display broad pathogen reactivity.” Scientific reports vol. 10,1 4611. 12 Mar. 2020, doi:10.1038/s41598-020-61632-9
- Hamadeh, R. M., et al. "Human natural anti-Gal IgG regulates alternative complement pathway activation on bacterial surfaces." The Journal of clinical investigation 89.4 (1992): 1223-1235.